

# Projections of changes in extreme storm surges for European coasts using statistical downscaling

Maialen Irazoqui Apecechea[1], Angélique Melet[1], Melisa Menendez[2], Hector Lobeto[2] and Jonathan B. Valle-Rodriguez[2]

[1]Mercator Ocean International, Toulouse, France
[2]IHCantabria - Instituto de Hidráulica Ambiental de la Universidad de Cantabria, Santander, Spain

*Correspondence to*: Maialen Irazoqui Apecechea (mirazoki@mercator-ocean.fr)

**Abstract.** Understanding future changes in extreme storm surges (ESSs) is critical for coastal risk assessment and adaptation. However, existing projections in Europe are often based on computationally expensive dynamical models, limiting ensemble sizes and thus confidence in projected changes. In this study, we develop a cost-effective statistical downscaling model (SDM) trained to replicate dynamically downscaled storm surges, enabling the generation of a pan-European ensemble of ESS projections based on 17 global climate models (GCMs)—substantially expanding previous efforts.

The SDM is trained on a storm surge hindcast and demonstrates stable skill across historical and future climates, effectively capturing projected ESSs changes given by dynamical simulations. Ensemble projections reveal robust multi-model mean (MMM) changes in the 10-year return level (RL10) of ESSs by 2100. Negative MMM changes are identified in the Mediterranean Sea (–7%), Moroccan Atlantic coast (–10%), and Danish Straits (–6%), while positive changes of around +6% are projected for the Celtic and Irish Seas, western Denmark, and the Gulf of Finland. Despite these robust signals, inter-model spread is substantial, with likely ranges (17th–83rd percentiles) extending from –25% to +17% across Europe, and changes of up to ±35% in individual models. The southern North Sea and northern Baltic Sea emerge as low-confidence regions, marked by particularly strong inter-model spread. Higher return levels (e.g., 100-year) show larger changes but increased uncertainty.

Our results underscore the importance of extended ensembles in projecting ESSs in Europe and demonstrate the value of statistical models for applications that demand extensive simulations—such as climate projections based on large ensembles, multi-scenario climate analyses, and detection and attribution studies—which can complement computationally expensive traditional dynamical downscaling methods.

## 1 Introduction

Extreme storm surges (ESSs), driven by intense wind forcing and low atmospheric pressure during storms, are a major contributor to coastal extreme sea levels and flood risk across Europe and many other regions globally (Woodworth et al., 2019). These events are particularly pronounced in regions with shallow depths and wide continental shelves—conditions typical of much of the northern European coastline. With ongoing climate change and associated mean sea-level rise, the



impacts of storm surges are expected to increase significantly. Even without changes in storm characteristics, rising mean sea levels will reduce the vertical buffer between the sea and the coast, dramatically increasing the frequency of today's high-impact events (e.g., Fox-Kemper et al., 2021). In addition, stormy conditions over the ocean and the induced storm

surge behaviour may also change under a warmer climate, further contributing to changes in future coastal hazards.

Despite the relevance of these processes, robust regional projections of future changes in storm surges remain limited, including for Europe. Most existing studies rely on hydrodynamic simulations to dynamically downscale (DD) climate model outputs. While physically detailed, these methods are computationally expensive, restricting ensemble sizes to a small number of global climate models (GCMs) and climate change scenarios (e.g., Chaigneau et al. 2024; Muis et al. 2020, 2022;

Vousdoukas et al. 2016). As a result, inter-model uncertainty assessment is limited, and the confidence in projected surge changes remains low.

In response to these limitations, statistical downscaling (SD) has emerged as a computationally efficient alternative. SD models aim to derive empirical relationships between large-scale fields (*predictors*) and local variables (*predictands*) —in our case, linking atmospheric fields to local storm surges. Several statistical methods exist with varying levels of complexity.

Multiple linear regression (MLR) techniques, which typically include a dimensionality reduction step of the predictors based on principal component analysis (PCA), remains widely used due to its simplicity, low computational cost, and high interpretability. Several studies have demonstrated its ability to reconstruct historical storm surges and wave conditions with skill comparable to, or exceeding, that of dynamical approaches (e.g., Cid et al., 2017; Harter et al., 2024; Tadesse et al., 2020). More complex methods based on Weather Types (WT) cluster synoptic atmospheric conditions into circulation

regimes and link them probabilistically to local surge responses (Anderson et al., 2019; Costa et al., 2020; Zhong et al., 2025), offering a physically interpretable, though less continuous, framework. More recently, neural network (NN) approaches are emerging as promising tools to statistically downscale marine variables. These methods can accommodate more flexible and complex predictor-predictand relationships, including non-linearities typically characterizing storm surges and their extremes. Recent studies (Bruneau et al., 2020; Tiggeloven et al., 2021) have demonstrated improved skill of NNs

to represent storm surge and their non-linearities at tide-gauges globally compared to MLR methods, with comparable performance to dynamical models, but a tendency to underpredict extremes persist. A very recent study (Hermans et al., 2025) showed that adapting the cost function to specifically target extreme events can alleviate this tendency of underprediction. While promising in terms of predictive accuracy, NN models require extensive tuning, large training datasets, and offer limited interpretability—making their application at large scale more challenging.

A key limitation of most SD approaches is their reliance on observed storm-surges in tide-gauge records for training, which restrict reconstructions to discrete coastal sites with sufficiently long and high-quality observations. Alternatively, hybrid downscaling methods have been developed, which use outputs from physically based numerical simulations as the predictand. This enables the training of SD models on spatially and temporally continuous storm surge fields, effectively replicating the behaviour of dynamical models at a fraction of the computational cost. Hybrid SD approaches have been



successfully applied to reconstruct historical storm surge fields at regional (Tausía et al., 2023) and global scales (Cid et al., 2017).

While widely used for past storm surge reconstructions, the application of (hybrid) SD to future storm surge projections remains limited. Only a few studies have explored this approach for regional projections, such as Cagigal et al. (2020) for New Zealand and Boumis et al. (2025) for Japan, but no such projections have been developed for Europe to date.

Furthermore, existing projection-focused applications have not evaluated whether the statistical relationships established under past (observed) conditions remain valid under forcing from climate models, nor have they explicitly assessed the assumption of stationarity in the predictor–predictand relationship between past and future periods. As such, the reliability of statistically downscaled storm surge projections under climate change remains under-explored.

In this study, we address these gaps by producing the first expanded multi-model ensemble (17 models) of pan-European storm surge projections using a hybrid statistical downscaling approach. We adopt a multi- linear regression framework for

its proved satisfactory performance in Europe (Tadesse et al., 2020) and its computational efficiency, which facilitates its scalability to the whole European coast. We train the model on reanalysis-forced hindcast simulations performed with a high-resolution dynamical downscaling storm surge model. The statistical model is adapted for application to global circulation models (GCMs) from the Coupled Model Intercomparison Project Phase 6(CMIP6), which present varying

spatial and temporal resolution. We validate the statistical model using an ensemble of four dynamically downscaled GCMs for historical and future climates. Finally, the trained model is applied to a 17-member CMIP6 ensemble to assess projected changes in extreme storm surges and their associated uncertainties across the European coastline.

## 2    Methods

### 2.1    Dynamical downscaling model (DDM)

The target storm surge hindcast (predictand) to be reproduced by the SDM is generated using the Regional Ocean Modelling System (ROMS) hydrodynamic model (Shchepetkin & McWilliams, 2005) in barotropic mode (2D). Based on the configuration developed by Cid et al. (2014), the model was implemented over a pan-European domain using an orthogonal grid, with a horizontal resolution ranging from 5 to 11 km, comprising a total of 272,382 grid points, and with bathymetry based on the ETOPO1 1 arc-minute dataset (Amante & Eakins, 2009). ERA5 reanalysis (Hersbach et al., 2020)

instantaneous hourly fields of meridional and zonal winds at 10-meters (U10,V10) and surface atmospheric pressure (PSL) were used as atmospheric forcing, and the inverse barometer effect was included as sea level open boundary conditions. For computational speedup, we thin coastal points by a factor of 10, leading to ~600 coastal points.



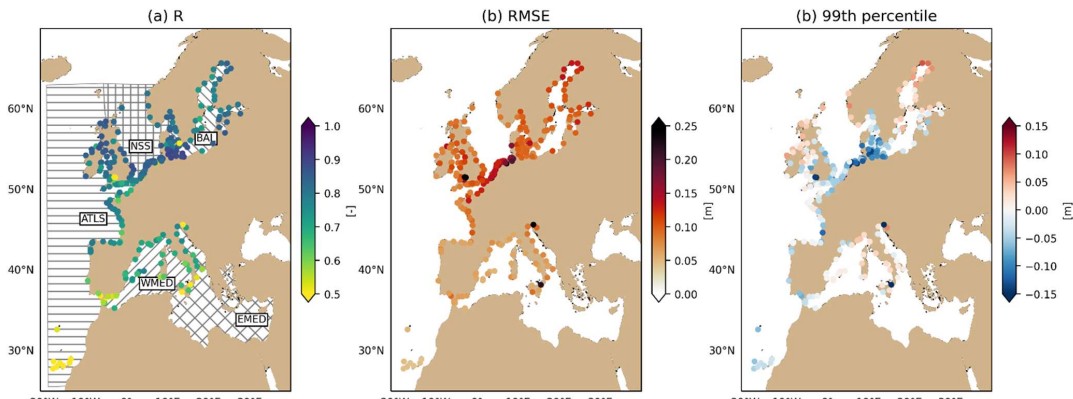

**Figure 1 Hydrodynamic model hindcast performance against GESLA3 tide-gauge observations (Haigh et al., 2023) for 1997-2015.**
**Left: RMSE. Right: Performance of the 10-year return level RL10. GESLA3 storm surge has been extracted after yearly tidal analysis (considering a minimum of 80% coverage for each year) and is computed relative to the yearly mean sea level (detrended). RL10 in GESLA3 is computed for stations with a minimum of 10 years of data in 1995-2015. EMED: Eastern Mediterranean Sea; WMED: Western Mediterranean Sea; BAL: Baltic Sea; NSS: North Sea; ATLS: Atlantic Shelf**

The DDM hindcast demonstrates satisfactory agreement with the storm surges observed in the GESLA3 tide-gauges (Fig 1;
see caption for processing details), yielding mean correlation (Fig 1-a). and RMSE (Fig 1-b) values of 0.76 and 10cm, respectively. Correlations are lower than average for the southern part of the domain, which probably reflects the contribution of baroclinic processes to the non-tidal residual in tide-gauges, which is not captured in the 2D barotropic model. RMSEs are higher than average in the North Sea (15cm), which might be expected given the larger storm surge amplitudes in the region. The DDM hindcast shows a general tendency for underpredicting high storm surges (mean -3 cm),
again most pronounced around the North Sea (-10cm, Fig 1-c). This underestimation of high storm surges is a well-documented limitation in similar hydrodynamic model simulations (Chaigneau et al., 2024; Fernández-Montblanc et al., 2020), which is likely attributable to inaccuracies in the representation of storm events in the atmospheric forcing data (Irazoqui et al. 2022). Nevertheless, as the primary aim of this study is to evaluate and apply statistical downscaling methods to replicate outputs from dynamical downscaling, this systematic underprediction relative to observations does not affect the
validity of our findings or conclusions.

In addition to the storm surge hindcast — which reproduces past storm surge conditions around European coastlines— we produce an ensemble of historical and future storm surge conditions by forcing the DDM with 4 CMIP6 models (Table 1). Future projections are based on an intensive fossil fueled socio-economic development with high future emissions (SSP5-8.5, Shared Socioeconomic Pathway 5, radiative forcing of 8.5 W/m2 by 2100, Meinshausen et al. 2020). The storm surge
datasets produced by the DDM are henceforth called *dynamically downscaled (DD)* storm surges.



**Table 1 Simulations performed with the dynamical downscaling model (DDM) and corresponding time coverage and forcing models. The global circulation models (GCMs) are part of CMIP6. Model specifications for the CMIP6 GCMs are presented in TableS1.**

| Epoch | Period | Forcing (U10, V10, PSL) | Temporal resolution |
|---|---|---|---|
| Hindcast | 1997-2021 | ERA5 reanalysis | Hourly instantaneous |
| Historical climate | 1995-2014 | MPI-ESM1-2-HR | 3-hourly instantaneous U10,V10 and |
| Future climate (SSP5-8.5 scenario) | 2015-2099 | EC-Earth3 CNRM-CM6-1-HR MRI-ESM2-0 | 6-hourly instantaneous PSL, except for 3-hourly instantaneous in MPI-ESM1-2-HR |

### 2.2 Statistical downscaling model (SDM)

Following our goal to study storm-surge extremes, we use multi-linear regression (MLR) to establish a relationship between the dynamically downscaled daily maxima storm-surge at each target coastal location in the DDM simulations (*predictand*) and daily aggregated forcing atmospheric fields (10-meter winds and mean sea level pressure) in a region of influence around each coastal point (*predictor*). We consider a temporal lag of up to two days between predictors and predictand, given the beneficial impact shown by Tadesse et al. (2020) in the European region. PCA (targeting a 99% of explained

variance) is employed to reduce the high dimensionality of the predictors to lighten the subsequent regression.

To enable the application of our SDM across CMIP6 forcings with varying spatial resolutions (Table S1), all atmospheric input fields—including ERA5 reanalysis—are remapped to a uniform spatial resolution of 1°. This preprocessing step ensures consistency in spatial resolution across training and application datasets. Notably, when trained on the hindcast simulation, the SDM learns to relate coarsened atmospheric predictors to storm surges generated using high-resolution ERA5

forcing (0.25°). Consequently, when applied to 1° CMIP6 predictors, the SDM effectively approximates the storm surges that would be expected under higher-resolution atmospheric conditions. Additionally, before applying the hindcast-trained SDM to CMIP6 forcings, the latter are bias-corrected relative to the ERA5 reanalysis by adjusting their mean and standard deviation at each grid cell over the reference period 1995-2014. This step crucially ensures compatibility in magnitude and variance between CMIP6 and ERA5 predictors, and hence a consistent projection of CMIP6 predictors onto the ERA5-based

principal components. The general workflow for the training and application of the SDM for climate projections is outlined in Fig 2.



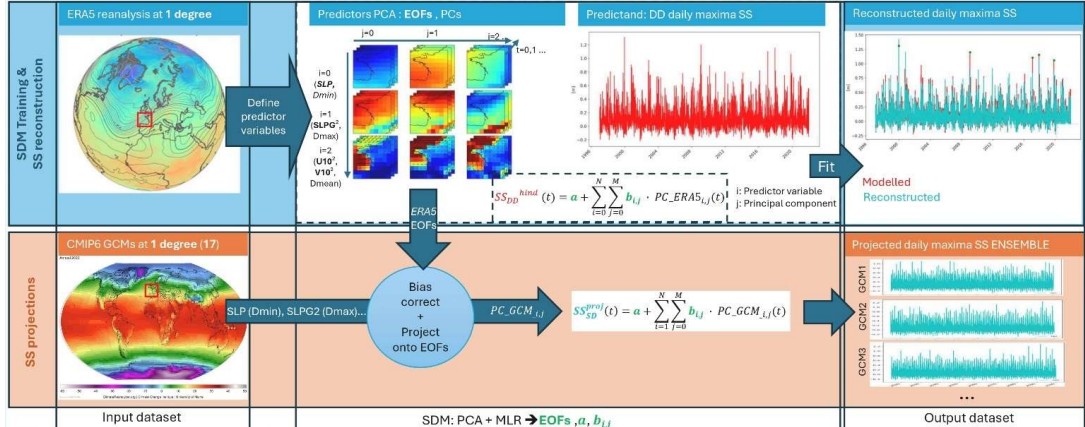

**Figure 2 Example of the workflow followed for the training and application of the statistical downscaling model (SDM) for the**
**reconstruction and projection of the dynamically downscaled (DD) daily maxima storm-surge (SS) at la Rochelle, France.**
**Predictor atmospheric variables (e.g. sea level pressure - SLP, sea level pressure gradient - SDPG, zonal and meridional wind**
**speeds at 10 meters - U10, V10) are further defined in Table 2. The SDM is trained on the hindcast simulation outputs forced by**
**the ERA5 reanalysis. The SDM is defined by the empirical orthogonal functions (EOF) — extracted through principal component**
**analysis (PCA) and representing the dominant modes of variability in the atmospheric fields around the target coastal point— and**
**the linear regression coefficients (a, $b_{ij}$) derived from multi-linear regression (MLR) between the principal component series**
**($PC\_ERA5_{i,j}$) and the target storm surge series ($SS_{DD}^{hind}$). Once these SDM elements are defined, global climate model (GCM)**
**atmospheric fields from CMIP6 interpolated at 1 degree are projected onto the EOFs ($PC\_GCM_{i,j}$) and combined through the**
**regression coefficients to produce storm surge projections ($SS_{SD}^{proj}$).**

### 2.3    Calibration of the SDM

First, a calibration phase is conducted to identify the optimal SDM configuration for the representation of daily maxima SS

along the European coastline, based on performance for the hindcast simulation. Different configuration choices and

predictor variables (Table 2) are tested. We aim for best overall performance at the least possible complexity (and

computational cost), indicated by the resulting number of principal components. In terms of predictors, we consider sets of

increasing complexity: we start from daily minima atmospheric pressure-SLP -representing the inverse barometer effect

(T1); we then add the daily maximum squared atmospheric pressure gradient-SLPG- as a proxy for geostrophic winds

(Rueda et al., 2016) (T2); finally, we account for the influence of zonal and meridional near-surface winds (U10, V10),

considering both daily mean and daily maximum values (T3 and T4, respectively), as well as their squared counterparts

(U10², V10²), represented by T5 (daily mean) and T6 (daily maxima). Squared wind components capture the nonlinear

effects associated with wind stress, which was shown to reduce biases on the statistical estimation of extreme storm-surges

compared to linear winds in Europe (Harter et al. 2024). We explore both daily maximum and mean wind speeds to reflect

different physical contributions to storm surge development—instantaneous wind stress for peak-driven surges, and





integrated wind forcing for more gradual surge buildup. As a last experiment, we explore an intermediate option between daily mean and maxima squared winds, where the mean over the 5 hours centered at the time of maximum winds is considered (T7). In terms of configuration choices, we test predictor fields in boxes of 3,6,9 and 12 degrees (D) centered at

each coastal point, and for 0,1and 2 days of lag (L) in the predictors. Lagged predictors allow the model to recognize multi-day storm dynamics, including slow-moving or pre-conditioning events that influence surge magnitude.

**Table 2 Options for the SDM configuration tested. Predictors include: sea-level pressure - SLP, sea level pressure gradient squared -SLPG, zonal and meridional winds at 10 meters -U10, V10- and corresponding squared values ($U10^2$,$V10^2$). Time aggregations: daily minima (Dmin), daily maxima (Dmax), daily mean (Dmean) and 5-hour mean around the time of maximum**

**daily wind (5hmean).**

| predictor variables | Bounding box size (degrees) | Time-lag (days) |
|---|---|---|
| SLP-Dmin(**T1**) | 3x3 (**D3**) | 0 (**L0**) |
| SLP-Dmin, SLPG-Dmax(**T2**) | 6x6(**D6**) | 1(**L1**) |
| SLP-Dmin, SLPG-Dmax, U10-Dmean,V10-Dmean(**T3**) | 9x9(**D9**) | 2(**L2**) |
| SLP-Dmin, SLPG-Dmax, U10-Dmax,V10-Dmax (**T4**) | 12x12(**D12**) | |
| SLP-Dmin, SLPG-Dmax, U10²-Dmean, V10²-Dmean (**T5**) | | |
| SLP-Dmin, SLPG-Dmax, U10²-Dmean, V10²-Dmean (**T6**) | | |
| SLP-Dmin, SLPG-Dmax, U10²-5hmean, V10²-5hmean (**T7**) | | |

We calibrate the SDM for the configurations summarized in Table 2, using ERA5 forcings and the hindcast data (1997-2021, 25 years). We evaluate performance through a k-fold cross validation, using 5 folds or splits (5 years), whereby the model is trained for k-1 folds at a time and tested on the remaining fold. The test folds, representative of the SDM performance for independent data, are finally combined into a complete time-series for 1997-2021. We evaluate standard metrics over the

whole time-series (root mean squared error -RMSE, Pearson correlation coefficient-R) as well as the mean bias (MB) for the tail of the distribution (> 99[th] percentile) to quantify performance for high storm surges.



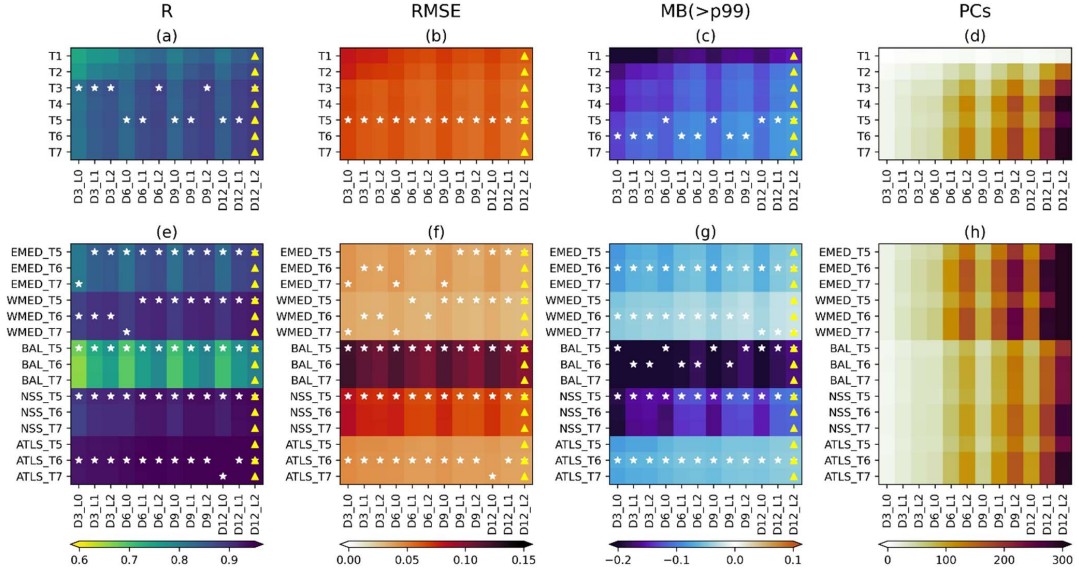

**Figure 3 Performance of the statistical downscaling model (SDM) for the reconstruction of storm-surges from the hindcast simulation for the different configuration choices in Table 2. Top - metrics averaged over Europe; bottom - metrics averaged over the following sub-regions: EMED: Eastern Mediterranean Sea; WMED: Western Mediterranean Sea; BAL: Baltic Sea; NSS: North Sea; ATLS: Atlantic Shelf. See Figure 1 for the definition of the geographical coverage of each region. Metrics are derived over the 1997-2021 statistically downscaled daily maxima storm surge, reconstructed piece-wise for data independent from training following a k-fold approach. R- Pearson correlation coefficient, RMSE-root mean square error, MB(>p99) – mean bias for the series above the 99th percentile. PCs – number of resulting principal components. For each performance metric, white _starts_ indicate best set of predictors (T) for each configuration (domain size-D, time lag -L, horizontal axis), and yellow _triangles_ best configuration for each set of predictors per domain (vertical axis).**

Average performance metrics across all coastal points (Fig 3a–c) indicate a general improvement in the SDM skill with increasing domain size (D), inclusion of temporal lags (L), and predictor complexity (T). Overall, the SDM achieves high correlations (>0.74) and low RMSE values (<0.08 m) across all configurations. However, for high storm surge values (Fig 3c), a systematic underprediction bias is observed.

Performance gains associated with larger D, longer L, and more complex T are attributed to both more informative predictors and a greater number of principal components (PCs) retained in the model (Fig 3d), which increase the degrees of freedom available for regression. This improved flexibility, however, comes at the cost of added computational complexity and a heightened risk of overfitting.

Certain configurations emerge as particularly beneficial. A domain size of at least 6° (D2) and the inclusion of a minimum 1-day lag (L1) significantly enhance performance across predictor sets and evaluation metrics. Predictor-wise, the addition of wind variables (T > 2) markedly improves correlation and RMSE. While performance under normal conditions appears



insensitive to further predictor complexity, high surge events show notable bias reductions when squared wind terms (T > 4) are included—consistent with findings by Harter et al., (2024). However, for T>4, no single predictor set emerges as the best-performing option for Europe as a whole and across configurations and metrics.

A regional decomposition of model skill using T>4 (Fig 3e–g, see Fig 1 for geographic regions) reveals strong spatial variability. Considering characteristic regional storm-surge variance (i.e., normalizing RMSE and MB), the SDM performs best along the Atlantic façade (ATLS), North Sea (NNS), and western Mediterranean (WMED), while performance is weaker in the eastern Mediterranean (EMED) and especially in the Baltic Sea (BAL). In these lower-performing regions, temporal lag proves critical; performance improves substantially with lags up to two days. Although increasing the domain beyond 9° yields minimal benefit, it significantly increases computational cost. Performance differences between T5 and T6 are negligible, favoring T5 due to its lower complexity.

Based on these findings, the configuration T5–D9–L2 is selected as the optimal model, balancing accuracy, complexity, and computational efficiency. This configuration yields average R, RMSE, MB values of 0.89, 5.5 cm, and 8.6 cm, respectively, which is comparable to other state-of-the-art data-driven reconstructions in Europe (e.g.,Tadesse et al., 2020). Time series comparisons (Fig 4) further demonstrate the skill of the SDM to accurately reproduce both average and extreme storm surge events.

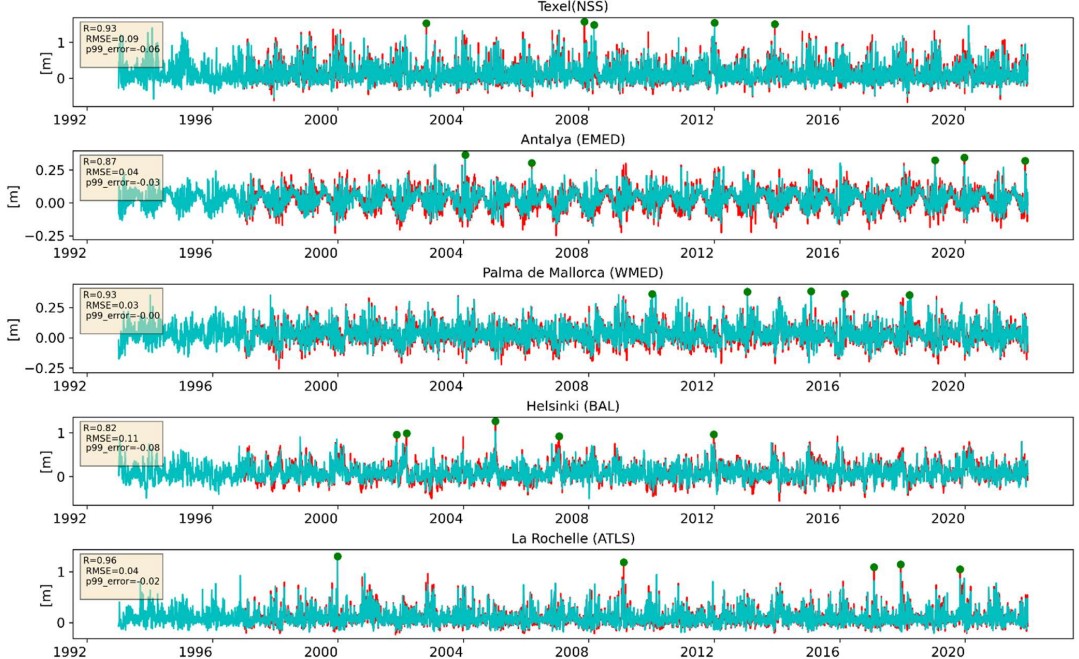



**Figure 4 Comparison between target coastal storm-surge from dynamical simulations (red, predictand) and the reconstruction using statistical downscaling (blue). The largest 5 events in the predictand are marked with a circle. Performance metrics added in boxes: R-Pearson correlation coefficient; RMSE- root mean square error; p99e – error in the 99th percentile. EMED: Eastern Mediterranean Sea; WMED: Western Mediterranean Sea; BAL: Baltic Sea; NSS: North Sea; ATLS: Atlantic Shelf. See Figure 1 for the definition of the geographical coverage of each region**

### 2.4 Extreme Value Analysis

Extreme storm surges in past and future climates are analyzed through stationary-on-slice extreme value analysis (EVA) using the package by Mentaschi et al., (2016). Assuming stationarity of SSs in 20/30-year time-slices, we fit a Generalized Pareto Distribution (GPD) on the SS peaks that exceed a threshold (known as the Peak Over Threshold method, POT). The selected threshold was based on an average rate of 3 extreme events per year in the considered time-slice (known as the Peak Over Threshold method, POT). Events are considered independent when separated by at least 3 days, considered the approximate time most storm events influence water levels at the coast (Wahl et al., 2017) and typically employed in extreme value analysis in Europe (Chaigneau et al., 2024; Haigh et al., 2016; Vousdoukas et al., 2016). We fit the GPD parameters using Maximum Likelihood Estimation.

### 3 Results

The calibrated SDM is trained on the hindcast simulation outputs (1997-2021) to establish the statistical relationship between predictors and daily maxima storm surges, which forms the basis for climate projections. Prior to application for climate projections, the suitability of the hindcast-trained SDM for GCM-based projections is assessed by comparing the corresponding reconstructed storm surges to those dynamically downscaled from the same GCM forcings for historical and future climates. In this validation, two crucial aspects are evaluated: the stationarity of the predictor-predictand relationship between historical and future periods, and the skill of the hindcast-trained SDM in approximating storm surges under climate model forcings. To our knowledge, this constitutes the first study to explicitly assess the extrapolability and robustness of a hindcast-trained SDM under climate change conditions, addressing a key limitation in previous approaches that apply such models without prior validation for SS projections. Finally, the validated SDM is applied to 17 CMIP6 models to generate a storm surge projection ensemble.

### 3.1 SDM validation for climate projections

To validate the SDM for climate projections, we inter-compare three storm-surge estimates: the dynamically downscaled estimates (*DD*), the storm surge reconstructions produced using the SDM trained on the historical DDM simulation outputs for each GCM (*SD*), and the storm surge reconstructions using the SDM trained solely on the hindcast simulation outputs (*SD_hind*). The *SD* reconstruction will inform on the differences in storm-surge projections incurred by the SDM model alone, while the *SD_hind* reconstructions will also incorporate differences stemming from the assumption that the statistical relationships derived from the hindcast (comprising principal components and regression coefficients) remain valid for





historical and future climate projections based on CMIP6 models. Notably, *SD_hind* allows to rely on a single dynamically downscaling simulation – the hindcast – as opposed to dynamical downscaling historical simulations for each GCM, as required for *SD,* which would result costly for large ensembles.

Since dynamical simulations are available for a limited 20-year period for historical climates, and hence for a fair
250    comparison between *SD* and *SD_hind*, we limit the SDM training to 20 years in all experiments exclusively for the validation: 1997-2016 when trained on the hindcast (*SD_hind*) and 1995-2014 when trained on the historical simulations (*SD)*. Reconstructions span the period 1995-2100. For validation, the periods 1995-2014 and 2080-2099 are analyzed for past (hindcast, historical) and future epochs respectively. Additionally, we focus on the 10-year return level (RL10) to reduce the influence of EVA errors on the validation.



### 3.1.1 Stationarity assumption

First, we assess the assumption of stationarity in the predictor–predictand relationship—namely, that the principal components and associated regression coefficients derived from past conditions remain valid under future climate scenarios. To evaluate this, we compare the skill of the SDM to reproduce storm surge climates in both historical and future periods when the model is trained exclusively on historical outputs of the DDM, either from the hindcast (*SD_hind*) or from each GCM-specific historical climate simulations (*SD*).

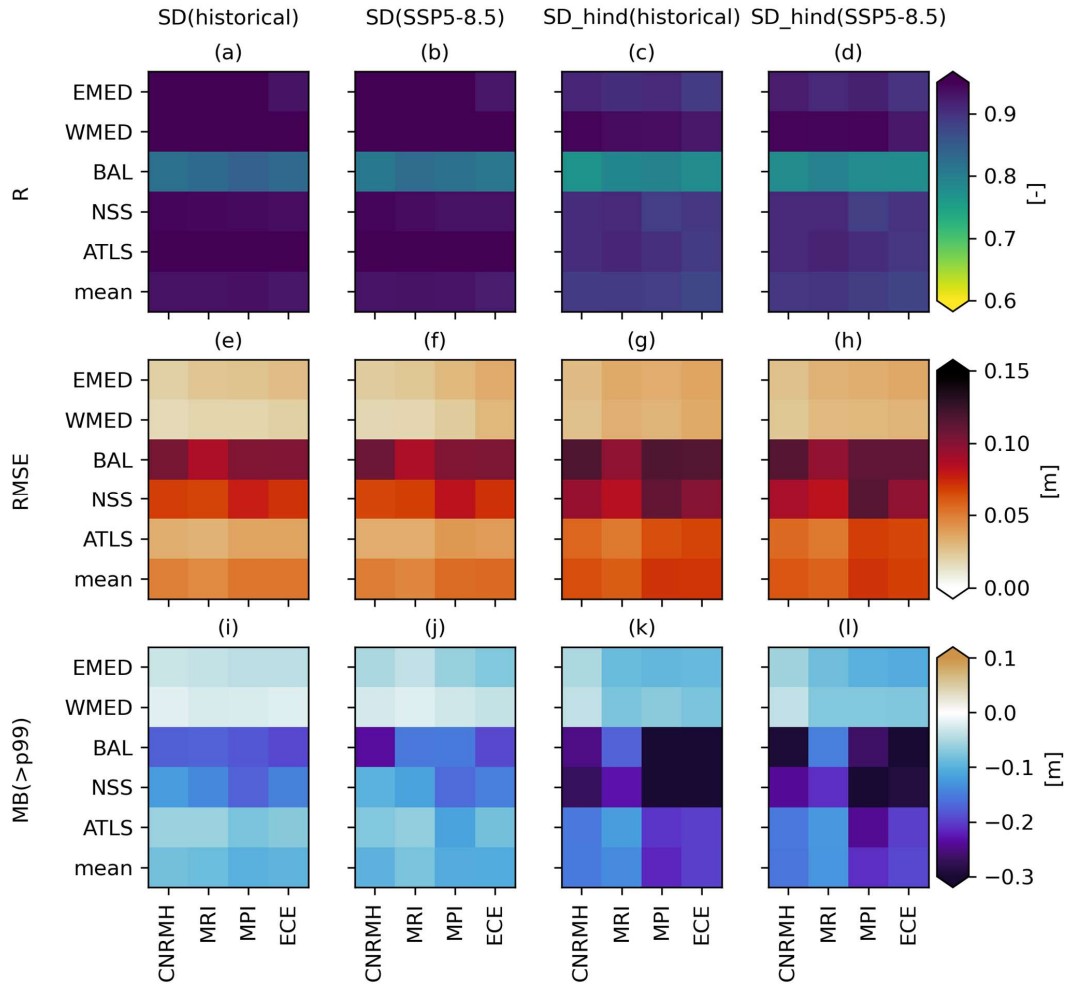



**Figure 5 SDM skill in the reproduction of storm surges for historical (1995-2014) and future (2080-2099, SSP5-8.5) climates for each GCM when trained on the historical dynamical simulations with each GCM (SD) and when trained on the hindcast simulation (SD_hind). The 4 GCMs are CNRM-CM6-1-HR (CNRMH), MRI-ESM2-0(MRI), MPI-ESM1-2-HR(MPI) and EC-Earth3 (ECE). EMED: Eastern Mediterranean Sea; WMED: Western Mediterranean Sea; BAL: Baltic Sea; NSS: North Sea; ATLS: Atlantic Shelf. See Figure 1 for the definition of the geographical coverage of each region.**

Regionally aggregated skill metrics (Fig 5) illustrate that performance metrics for SDM trained on each GCM-specific historical simulations (*SD*) are very comparable to those seen for the hindcast during calibration (Fig 3). *SD_hind* performs generally similar to *SD*, although it exhibits marginally reduced performance in reproducing the *DD* storm surges of climate simulations, which is consistent with the fact that GCM-specific information was not included in the *SD_hind* training process. Particularly, the underprediction of high storm surges (>99th percentile, Fig 5i-l) is more pronounced for *SD_hind*. Nevertheless, when comparing performance metrics between historical and future periods, a remarkably strong stationarity is found for both *SD* and *SD_hind* and across performance metrics. For the tail of storm surge distributions (Fig 5i-l), minor differences between periods are found for certain models and regions (e.g. Baltic Sea for CNRM-CM6-1-HR), but overall performances remain virtually unchanged. These results support the validity of applying SDMs trained on past conditions to reconstruct future climates.

### 3.1.2    Future ESS changes

Evaluating the ability of the SDM to capture projected *changes* in ESSs is central to its application in climate impact studies. Projecting relative changes between historical and future periods and then adjusting observational or reanalysis baselines accordingly is commonly done in climate science to minimize the influence of biases in the GCMs in future projections. The focus here is therefore assessing whether the SDM, trained under present-day conditions (represented by the hindcast), can reproduce the climate change signals in ESSs simulated by GCM-forced dynamical downscaling (*DD*).




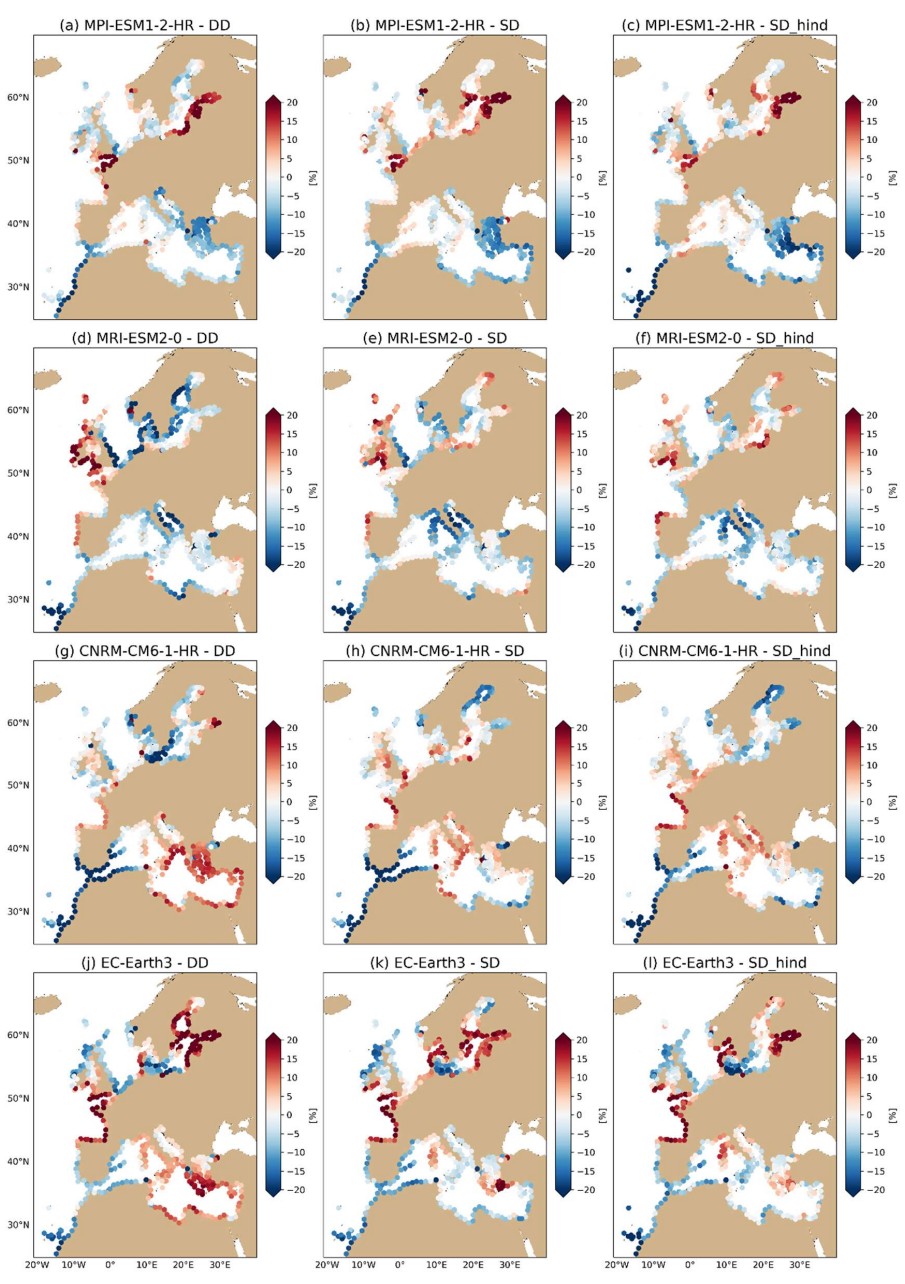



**Figure 6 projected changes (%, extreme value analysis on [2080-2099] vs [1995-2014]) in the 1 in 10 year storm surge event (RL10) for dynamical climate simulations (*DD*, left), statistical reconstructions trained on each historical climate simulation (*SD*, middle) and statistical reconstructions trained on the hindcast forced by ERA5 (*SD_hind*, right). For a fair comparison between *SD* and *SD_hind*, both are trained on 20-yr periods (1995-2014 and 1997-2016 respectively).**

Changes in storm surge RL10 based on dynamical projections reveal considerable inter-model spread, with regional changes reaching ±20% (Fig 6-a,d,g,j). Statistical projections trained independently on each GCM (*SD*) replicate the main spatial

features of the *DD* projections, demonstrating the SDM's skill to replicate GCM-specific climate responses. However, the magnitude of the projected changes appears slightly reduced in some regions. Projected changes given by certain models for the eastern Mediterranean Sea (CNRM-CM6-1-HR and EC-Earth3) and the Baltic Sea (MRI-ESM2-0 and CNRM-CM6-1-HR) are not fully captured in the GCM-specific SDM (*SD*). These might reflect the lower performance of the SDM identified for these regions during calibration (Fig 3).

When using the SDM trained solely on the hindcast (*SD_hind*), spatial patterns and relative amplitudes of RL10 change are generally well preserved across GCMs. This supports the applicability of the *SD_hind* setup for climate projections, with the added advantage of requiring a single simulation for training (the hindcast). However, differences can be notable for some coastal sections, which might be explained by the fact that ERA5-based PCs do not always fully explain GCM predictor variability for specific models and regions. As such, *SD_hind* reconstructions can only account for future storm surge

changes linked to the identified ERA5-PCs, and not to novel atmospheric conditions or different modes of variability that may be present in GCMs. A comparison of the historical RL10 performance for the different downscaling methods (*DD, SD, SD_hind*) relative to the hindcast (Fig S1) supports this argument: While *DD* results show widespread overprediction of RL10 across GCMs, applying the *SD_hind* model results in performance closely matching that of the SDM-based hindcast reconstruction for all GCMs. This is attributed to the combined effect of bias-correction on GCM fields and the use of

ERA5-based PCs.

Overall, the *SD_hind* model captures the projected patterns and magnitudes of change with reasonable accuracy across most regions and GCMs. While some limitations remain—particularly in regions where predictor–predictand relationships are weaker or GCM atmospheric variability differs from ERA5—the results support the broader applicability of hindcast-trained SDMs for climate projections of storm surge extremes along European coastlines.

### 3.2 Statistical ensemble projections

Following validation, the SDM trained on the full hindcast (1997-2021) is applied to generate an unprecedented ensemble of storm surge projections spanning the 21st century (1970–2100), based on 17 CMIP6 models (see Table S1 for model details). The 17 models are selected based on the availability of high-resolution atmospheric forcing— 3-hourly instantaneous wind fields and 6-hourly instantaneous mean sea-level pressure —for both the historical period and the SSP5-

8.5 scenario. Besides more comprehensive multi-model means, the expanded ensemble size enables a quantitative assessment of the robustness of projected changes in ESSs, measured by the fraction of models that agree on the sign of change. We define robust changes when 13/17 models agree in sign (closest estimate to the 80% ratio used in IPCC AR6).



Additionally, it allows, for the first time, the estimation of a likely range of ESS changes (represented by the 17th and 83rd percentiles) in line with IPCC uncertainty language.

For the 2050 time horizon (Fig 7a), the ensemble ESS projections show minor absolute multi-model mean (MMM) changes in RL10 across most European coastlines (<5%), except for sections of the Alboran sea and the Atlantic Moroccan coast (changes down to -11%). However, the likely range spans [-19, 11]% across Europe, with upper estimates (83rd percentile) reaching > 10% in the Celtic Sea, Gulf of Finland and western Danish coast, and lower estimates (17th percentile) showing widespread negative changes in RL10 in both northern (-6% average) and southern Europe (-10% average) (Fig 7c,d, see

also Fig S3 for individual ensemble members). The Mediterranean Sea is the only region where a consistent signal emerges, with a substantial fraction of models projecting a decrease in RL10 values (Fig 7f).

By the end of the century (Fig 7e), more pronounced MMM RL10 changes are projected with a spatial pattern that appears to scale with the mid-century changes, and widespread regions of substantial inter-model agreement emerge (Fig 7f). These two features suggest that changes in RL10 are likely (and at least partly) driven by a forced response to anthropogenic climate

change. A robust (13/17 models) reduction in RL10 is projected along the Mediterranean coasts (means across western and easter sections of -8 and -6%, respectively), the Atlantic façade of the domain south of 45°N (mean -10%), and around the Danish Straits (mean -6%). In contrast, a robust increase is projected for the coasts around the Celtic and Irish seas, western Denmark and the Gulf of Finland, with average changes in those regions of around +6%. However, the inter-model spread proves large across Europe, with upper range estimates (83rd percentile) reaching >10% in most of northern Europe and

lower range estimates (17th percentile) yielding values < -13% across southern Europe, reaching -25% around the Moroccan and Canarian Atlantic coasts. Notably, individual models may project changes of up to ±35% (Fig S4). While small MMM changes are projected for the Bay of Biscay, eastern UK, southern North Sea and northern Baltic Sea, the latter two regions display the widest likely ranges in Europe, reflecting very low confidence in projected regional RL10 changes.

The regions where robust changes have been identified broadly agree in sign with previous literature on projected changes of

SS RL10, despite different models being employed (Makris et al., 2023; Muis et al., 2022; Vousdoukas et al., 2016). However, the exact extents and magnitudes may differ substantially. For example, Muis et al. (2022) and Vousdoukas et al. (2016) identify positive future ESS changes across the Baltic Sea, while in our results substantial positive changes are limited to the eastern Baltic Sea. These studies also identify regions with substantial signals which are not emerging in our ensemble (e.g. the south-eastern North Sea in Vousdoukas et al. (2016), which may result from the use of smaller ensembles

which underrepresent inter-model variance in storm-surge projections. In contrast, the widespread reduction in future ESSs throughout the Mediterranean Sea identified in our ensemble are consistent across studies.

For RL100 (Fig 7i-p), which represents rarer and more extreme events, the projected spatial pattern of change generally mirrors that of RL10, with similar regional "hotspots" across Europe. For 2050, magnitudes are comparable to those for RL10, but for 2100 projected changes are generally larger for RL100, particularly on regions showing an increase in return

levels – Celtic Sea (+8.4%), wester Danish coast (+8.8%), Gulf of Finland (+9%). Exceptionally, projected decreases in RL100 are milder than those for RL10 in the Mediterranean Sea (mean -3.5% vs -6.3%) and Atlantic façade south of 45N (-



6% vs -10%). The spatial pattern also appears to scale between middle and end of the century. However, the level of agreement among models is substantially lower than for RL10 and shows a minor increase between periods (Fig 7i vs m). The inter-model spread is substantially larger than for RL10, resulting in very wide likely ranges across Europe (Fig 7k,l,o,p,

see Fig S5 and Fig S6 for individual ensemble members). This reduced agreement for high return periods is likely driven by uncertainties in the estimation of the shape parameter, which governs the tail behavior of the distribution. Changes in this parameter between historical and future periods are difficult to constrain due to limited data on high return period events, resulting in high inter-model variability. When the shape parameter is held constant between historical and future periods for each model (Fig S7), the inter-model agreement in projected RL100 changes improves considerably, and the resulting

change magnitudes become more comparable to those of RL10. This suggests that much of the disagreement in RL100 projections stems from uncertainty in the EVA, and particularly in estimating how the shape parameter evolves with climate change.







**Figure 7 Muti-model mean (MMM) projected changes [%] in the 1 in 10 year (RL10, a,e) and 1 in 100 year (RL100, i,m) storm surge return levels by middle (a,i) and end (e,m)-of the 21$^{st}$ century generated by the hindcast-trained statistical downscaling model (SDM, training period 1997-2021). Corresponding ratio of models agreeing in the sign of projected changes (second column), 83$^{rd}$ (third column) and17th (fourth columns) percentiles, indicating the likely range as per IPCC definitions. Extreme**





value analysis is computed for 30 year periods: baseline [1895-2014], middle of the century [2035-2064], and end of the century [2070-2100]. For reference, in a 17-model ensemble, the 17$^{th}$ and 83$^{rd}$ quantiles correspond to the lower/higher ~3 models. The % model agreement represents confidence in the sign of projected changes. Those with ratio >80% (here, >=13/17 models) are marked with white circle edges in panels b,f,j,n.

Given the substantial spread exhibited by the 17-model ensemble, using a smaller subset of models—as commonly done in previous studies—may result in biased estimates of future storm surge changes and notably of their associated robustness. To illustrate this sensitivity, we compute the multi-model mean (MMM) end-of-century RL10 changes from ensembles of varying sizes (ranging from 3 to 16 models), constructed through random sampling within the full 17-model ensemble for a set selection of coastal locations where the full ensemble (N=17) projects robust RL10 changes (agreement in sign >~80%)( Fig 8).

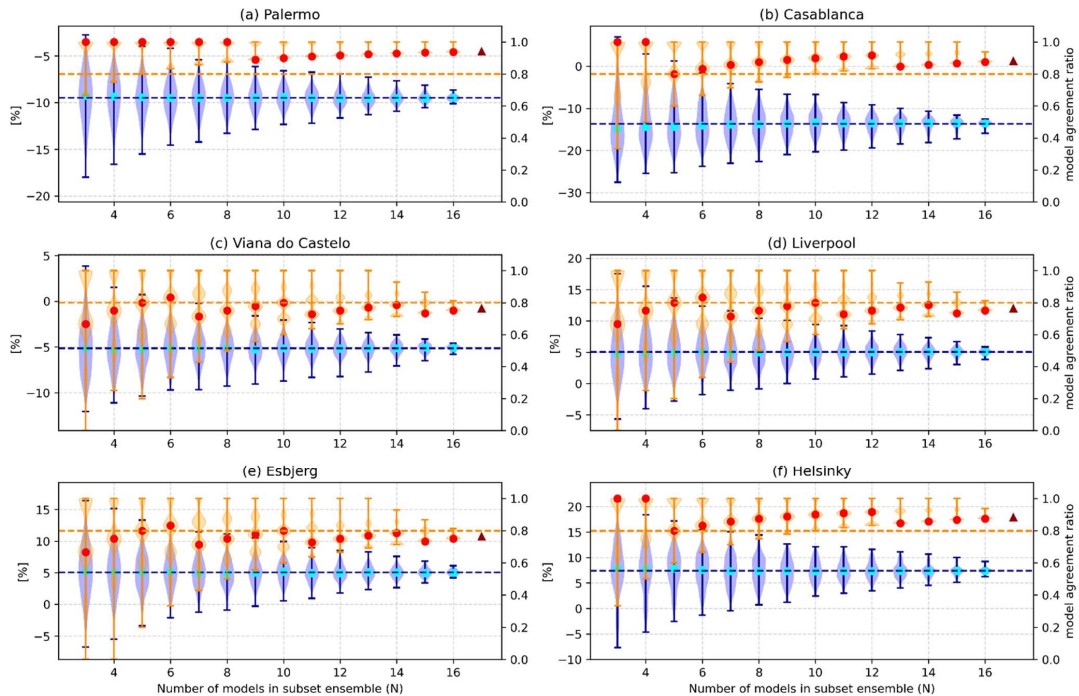

**Figure 8 Evolution of the multi-model mean (MMM) projected changes of the 1 in 10 year storm surge return level (RL10, blue) and corresponding ratio of models agreeing in sign (orange) for an increasing number of ensemble members (N) for a subset of coastal locations throughout Europe, using projections given by the hindcast – trained statistical downscaling model (SDM). For each N, N models in the ensemble are randomly sampled for a maximum of 2000 iterations, and the resulting distributions of MMMs and inter-model agreement ratio are plotted using violin plots. Dots indicate medians, and whiskers indicate the maxima/minima values of the distributions. The horizontal blue line indicates the MMM changes for the full 17-model ensemble, and the dark red triangle indicates the corresponding ratio of models agreeing in sign. The orange dashed line indicates the 80% model agreement ratio, used in IPCC assessments to assess robustness of projections. By construction, this metric can only attain**



**discrete levels that depend on N (=1/N). The most frequently attained discrete levels for each N are represented by the clusters in the violin plot.**

The convergence behavior of projected changes varies considerably across stations, with substantial spread in the multi-model mean (MMM) across different random sub-ensembles, particularly for ensemble sizes N<7. At N=3, the MMM can differ by up to ~30%. For instance, in Esbjerg and Liverpool, projected changes in RL10 range from -5% to +18%, corresponding to maximum absolute differences of 53 cm and 27 cm, respectively, between sub-ensembles.

The influence of ensemble size on the robustness of projections—defined as the fraction of models agreeing on the sign of change—varies markedly by location. In Casablanca and Helsinki, despite large variability in MMM values, more than 50% of sub-ensembles yield robust changes (agreement ratio > 0.8) even at small ensemble sizes, with robustness achieved unanimously for all sub-ensembles at N≥8. In Palermo, convergence is even more rapid, with high agreement (ratio > 0.7) reached from N>=4.In contrast, locations such as Esbjerg, Viana do Castelo, and Liverpool exhibit less robust behavior:

while the full ensemble shows agreement near the 0.8 threshold, sub-ensemble agreement diverges rapidly with smaller N. In some cases (e.g., N=3 or 4), sub-ensembles may project robust changes of opposite sign relative to the full ensemble (indicated by agreement ratios dropping below 0.2). These findings underscore the risk of drawing misleading conclusions from small ensemble sizes and highlight the importance of ensemble size in ensuring accurate quantification of uncertainty in future storm surge projections.

## 4    Discussion and perspectives

The SDM developed in this study has shown satisfactory skill to reproduce ESS changes projected by dynamical downscaling models, demonstrating the potential of the current hybrid approach to perform storm surge projections that would otherwise be very computationally demanding. However, several limitations and assumptions apply to our approach.

Regarding the statistical downscaling approach chosen, we have shown that MLR leads to a systematic underprediction of
the target (predictand) extremes, despite achieving very satisfactory performance for normal conditions. While our results have shown that this negative bias has a limited impact on broad-scale projections of ESSs changes, locally the differences can be substantial, and may amplify for higher return periods. In the standard PCA used in our SDM, all grid points contribute equally to the decomposition, regardless of their relevance to the target coastal site. Future work could explore distance-weighted EOF extraction (Baldwin et al., 2009), which emphasizes atmospheric variability near the site of interest,
potentially yielding principal components more representative of local storm surge drivers. Other more complex data-driven approaches than MLR, such as weather types (Costa et al., 2020) and neural networks targeted to extremes (Hermans et al., 2025) have demonstrated potential for a more efficient extraction of the atmospheric features that drive ESSs, but their use at regional to continental scale has not been proven to date. The vast majority of studies employing data-driven approaches for storm surges have either not assessed extreme events or have declared a tendency to underpredict them (Bruneau et al., 2020;
Tiggeloven et al., 2021). In this regard, a key shortcoming of employing a hybrid downscaling approach (as opposed to



targeting observed storm surges) is that the skill of the SDM will be strongly conditioned by the skill of the dynamical model. In this study, as well as in several previous ones, a tendency of the storm surge hindcast to underpredict ESSs has been identified (Fernández-Montblanc et al., 2020; Irazoqui Apecechea et al., 2023). For future projections, the reliability of storm surge estimates—whether derived from dynamical downscaling or statistical downscaling —ultimately depends on the

skill of the forcing GCM. Our results indicate that GCM-driven biases in ESSs persist, even after the bias correction applied in the forcings for the SDM reconstructions. Beyond advanced bias correction methods, alternative approaches to account for GCM fidelity include weighting the multi-model mean based on each GCM's ability to reproduce relevant European atmospheric patterns (e.g., via weather types, Borato et al. 2024; Cagigal et al. 2020) or on a site-specific basis, using each GCM's historical skill in reproducing ESSs at coastal locations (e.g., Fig S2).

A key assumption in the current approach—and in data-driven methods more broadly—is that the dominant predictors identified during training (in our SDM, the leading principal components derived from the 25-year ERA5 reanalysis) adequately capture the spectrum of atmospheric variability. However, given the presence of decadal to multi-decadal variability in atmospheric circulation patterns (Laurila et al., 2021), it is possible that some low-frequency modes are underrepresented or missing from this relatively short training period. Moreover, rare but high-impact storm events may be

under sampled. Nevertheless, the stationarity tests in Sect 3.1.1 suggest that the extracted principal components are sufficiently robust across epochs, and that low-frequency internal variability has a limited impact on the SDM skill.

In the same vein, applying the hindcast-trained SDM to both historical and future climate simulations effectively filters out storm surge events associated with novel atmospheric states or modes of variability that may be present in the GCMs but not in the hindcast. This filtering effect is further illustrated by the explained variance analysis after projecting GCM fields onto

hindcast-based EOFs (Fig S9), which reveals generally strong representativity across Europe, though with notable reductions for certain models and regions (e.g. the Mediterranean Sea for HadGEM3-GC31-MM). While such filtering may be justified for historical periods—serving to exclude potentially unrealistic conditions simulated by GCMs (assuming limited influence from internal variability)— its application to future climates is more problematic, as novel atmospheric conditions or variability modes may emerge. While the main patterns of projected RL10 changes were well captured by the SDM

compared to dynamical downscaling (Fig 6), this filtering effect of the PCs might explain some of the differences found locally between the two estimates. Finally, an additional contributor to limitations in the skill of the hindcast-trained SDM to reproduce DD storm surges for climate projections may be the mismatches between the ERA5 and GCM land-sea masks. This mismatch may affect the weighting assigned to winds in grid cells adjacent to the coastal target locations during projection onto the ERA5 EOFs.

The use of 1-degree resolution atmospheric predictors might further impact the skill of the SDM in reproducing dynamical simulations that are forced by the reanalysis and GCMs at their original, often higher than 1 degree resolution. While this step was needed to render a SDM compatible across forcings, studies suggest this resolution remains adequate: Agulles et al. (2024) found storm surges and extremes are well represented at 1-degree atmospheric forcings over Europe, and Costa et al. (2020) showed that such degraded predictors best preserved variability during dimensionality reduction for La Rochelle.



Indeed, negligible impacts on the SDM skill were found when using the original, high- resolution (0.25°) ERA5 predictors across Europe, except for occasional extreme events in selected locations (not shown). As downscaled CMIP6 datasets for Europe (e.g., Euro-CORDEX[1]) become increasingly available, the potential added value of higher-resolution atmospheric forcing—more closely aligned with ERA5's native resolution—for statistical storm surge projections warrants further investigation. Alternatively, storm surge simulations could be dynamically downscaled for a limited (e.g. 25 year) historical

slice from each GCMs. These would serve to train GCM-specific SDMs at native resolution, which could then be applied to future projections. This approach offers a middle ground in computational cost between the current method and full dynamical downscaling of the ensemble, while avoiding the potential limitations of extrapolating ERA5 predictor modes to GCMs.

We have shown that relying on small ensembles to assess future extreme storm surges—an approach commonly adopted in

the literature using hydrodynamic models—can lead to low-confidence estimates of both the magnitude and direction of projected changes. Importantly, our results highlight that the decision to either fix or allow the shape parameter of extreme value distributions to vary in future periods has a substantial influence on the projected magnitude and robustness of high return levels, such as RL100. There is currently no standardized practice in climate impact studies regarding this choice: some studies fix the shape parameter across time periods (Mentaschi et al., 2016; Vousdoukas et al., 2017) while others

allow it to evolve (Muis et al., 2022). This underscores the need for clearer methodological guidance and more systematic sensitivity analyses in future storm surge projection studies.

Despite the expanded ensemble size employed in this study, projections of future changes in ESSs still exhibit substantial spread. While part of this spread may stem from differing forced responses across GCMs, internal variability in storm surge extremes may play a significant role. Recent studies have highlighted the presence of multi-decadal variability in observed

extremes (Cheynel et al., 2025), with strong links to large-scale climate modes such as the Arctic Oscillation (Lobeto & Menendez, 2024). Consequently, not only may internal climate variability contribute to future ESS changes, but differing phasing of this variability across GCMs may further amplify inter-model spread. This internal variability presents a major challenge for the detection and attribution of changes in extreme storm surges. Non-stationary extreme value analysis (e.g. Lobeto & Menendez, 2024; Mentaschi et al., 2016) offers a potential pathway to disentangle internal variability from

externally forced trends by incorporating covariates—such as dominant climate modes—into the distribution parameters. Analyzing the residual trends in the extreme value distribution after accounting for these modes could improve the robustness of projected ESS changes. In this context, cost-effective statistical downscaling models like the one developed in this study offer significant value for detection and attribution efforts. They enable the generation and analysis of long storm surge time series under fixed pre-industrial radiative forcing (i.e. using hundreds of years of *piControl* GCM forcings, upon

availability of predictors at high temporal resolutions), which are essential for characterizing the influence of internal

---

[1] https://www.euro-cordex.net/



variability but are highly costly to produce with classic dynamical downscaling. Expanding our projections to additional SSP scenarios could also help in the interpretation of the projected changes.

## 5    Conclusions

In this study, we have developed and applied a cost-effective statistical downscaling model (SDM) to generate the first expanded, pan-European ensemble of storm surge projections based on 17 CMIP6 models. This significantly extends the ensemble size compared to previous regional assessments, which have typically relied on computationally expensive dynamical downscaling approaches that limit the feasible number of models.

Our SDM is trained to replicate dynamically downscaled storm surge outputs, enabling the reconstruction of spatially and temporally seamless surge estimates across the European coastline. We evaluated the performance of the SDM, which was

trained on ERA5 reanalysis and historical storm surge estimates, in replicating projections derived from dynamically downscaled simulations. The model demonstrated stable skill across both historical and future climates. While it tended to underestimate extreme storm surges (ESSs) during the historical period, it was able to capture projected changes in ESSs consistent with those produced by dynamical simulations.

The resulting ensemble projections reveal negligible changes in 10-year return level ESSs by mid-century (2050), but robust

changes (defined as agreement in sign across ≥13 of 17 models) emerge by the end of the century in several regions. Robust negative changes are projected for the Mediterranean Sea (–7%), Moroccan Atlantic coast (–10%), and Danish Straits (–6%), while positive changes of around +6% are projected for the Celtic and Irish Seas, western Denmark, and the Gulf of Finland. Despite these identified regions of robust changes, multi-model mean (MMM) changes are generally modest ([-15,8] % across Europe) but the likely ranges—defined by the 17th and 83rd percentiles—are wide, reflecting substantial inter-model

spread in projected changes of ESS 10-year return levels. In some regions, individual models project changes as large as ±35%. For the southern North Sea and northern Baltic Sea, our results reveal low confidence in projections of ESS changes, given by the combination of pronounced inter-model spread and low inter-model agreement on the sign of projected changes.

For higher return levels (e.g., RL100), there is considerably less model agreement on the sign of the projected changes and a

strong inter-model spread, driven in part by uncertainties related to future changes in the shape parameter of the extreme value distributions.

Our findings underscore the importance of using large ensembles when assessing future changes in extreme storm surges, as small ensemble sizes can lead to low confidence estimates. Future research should aim to identify and quantify the sources of spread in ESS projections, particularly the role of internal variability in extremes. In this regard, cost-effective statistical

models such as the one developed here provide a powerful tool for advancing detection and attribution studies, by enabling efficient production of long-term storm surge reconstructions across multiple GCMs, scenarios, and regions



## 6    Code availability

The Regional Ocean Modelling System (ROMS) code used for the dynamical dowsncaling of storm surge is freely accesible through the ROMS website (https://www.myroms.org/index.php ).

## 7    Data availability

Data on projected changes on extreme storm surge extremes using statistical downscaling are available on request. The tide gauge data used for validation are available on the GESLA website (at http://www.gesla.org, Haigh et al., 2023). The atmospheric fields from the 17 CMIP6 GCMs used to force the dynamical and statistical downscaling models are freely accessible via the different nodes attached to the ESGF server, such as https://esgf-node.ipsl.upmc.fr/ (ScenarioMIP dataset, historical and ssp585 experiments).

## 8    Author contribution

MIA, AM and MM designed the scope of the study and experiments. JBV performed the dynamical downscaling simulations. MIA, MM and HL designed the statistical downscaling model and calibration process. MIA carried out all statistical downscaling experiments and analyses in the manuscript and prepared the manuscript with contributions from all co-authors.

## 9    Competing interests

The authors declare that they have no conflict of interest

## 10    Acknowledgements

The authors are grateful to Lorenzo Mentaschi for providing the code used to perform the extreme value analyses, and to Alisee Chaigneau for her guidance on using the package. We also thank Adrian Acevedo Garcia for organising the transfer of the dynamical simulation outputs. Some parts of this manuscript have been rephrased using AI tools (e.g., ChatGPT), based solely on content originally written by the authors.

## 11    Financial support

This study has been accomplished within the Coastal Climate Core Services (CoCliCo) project, funded by the European Union's Horizon 2020 research and innovation programme under grant agreement No 101003598.



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
