# Peer review of "Projections of changes in extreme storm surges for European coasts using statistical downscaling"

_EGUsphere, 2025_

## Referee Comment (RC1)

This manuscript presents a statistical reconstruction of storm surge records using 17 climate model projections along the European coastlines. It is the result of a significant computational effort, combining a (relatively) small set of dynamical numerical simulations and a data-driven model based on multiple linear regression. The methods are well described and sound. All the details are provided for the calibration and choices of the statistical model parameters. However, I have some concerns on the application of the model, and the presentation and interpretation of the results. I think the manuscript requires a major revision before being suitable for publication.

One major concern is the focus on extreme storm surges. The results of the statistical model applied to ERA5 forcing fields are daily records of storm surges that, when compared with the benchmark results of the dynamic simulations, provide satisfactory performance in terms of averaged storm surges. This is shown in Figures 3 and 4 that show the capabilities of the statistical method in terms of correlation and RMSE. The only metric focusing partly on extremes is the bias of the 99th percentile, but this is not representative of the ability of the statistical records to capture extreme events and reproduce return levels, which are the results that are analysed later on. In fact, the comparisons in figure 4 show that discrepancies can be quite large for individual events. This is something wellknown for the statistical model based on Tadesse et al (2020). The approach is good in simulating the mean storm surge climate but displays limited accuracy with the extremes, and this is a major shortcoming that must be reflected in the manuscript. I attach below an example for a tide gauge in Brest that I produced some time ago for an assessment of data-driven models. Although the differences with the dynamic simulation are expected to be smaller than in this example, as the extremes will be also underestimated (shown e.g. in Figure 1), it is clear that the statistical model is not particularly well suited for extreme storm surges. I am not suggesting that the authors should change the statistical approach, I believe it has its value. I think, though, that the ability of the method needs to be better described, particularly concerning extremes. To do so, I suggest using qq-plots instead of time series to evaluate model performance. Likewise, mapping the differences in maxima or yearly maxima and/or return levels between statistical and dynamical approaches forced by ERA5 would provide the required information to the reader.

A second major concern is related to the discrepancies between dynamical and statistical simulations in climate models for some particular regions. As shown in Figure 5, regions as the Mediterranean (CNRM-CM6-1-HR) and the Baltic (MPI-ESM1-2-HR) indicate opposite changes in projected storm surges using dynamical and statistical models. To a lesser extent, also the western of the British Isles and the southern North Sea display different patterns. This needs to be clearly described. I do not think that the patterns are similar and only the magnitudes change, as claimed the lines 289-290. I agree, though, that using the hindcast instead of historical simulations for the training is mostly fine.

These discrepancies hinder the interpretability of the projected storm surges presented in figure 7. The regions where the statistical and dynamical models clearly differ should not be discussed or even mapped. This includes the Baltic and the Eastern Mediterranean Seas (the western Mediterranean seems consistent in the validation). In addition, the uncertainty range is very high for the 100-year return levels, ranging from negative to positive values. This indicates that there is no confidence in the multimodel ensemble means (panels 7i and 7m). I suggest focussing only in the 10-year return level. This is also consistent with the fact that the statistical model is less reliable for the most extreme events.

Another issue that I think requires some attention is the discussion about long-term variability in the atmospheric patterns and its impact on the performacen of the statistical model (lines 430-449). I do not think that the long-term modulation of low-frequency climate modes affects the results of the statistical model. Storm surges are caused by synoptic systems. These can be altered in frequency and magnitude by large-scale climate modes. However, the synoptic systems are still the same. In other words, changes in large-scale atmospheric conditions, like more blocking patterns, shifts of NAO, etc, will modulate the frequency and the intensity of the systems that generate the storm surges, but will not change the

process and the type of system, nor the response of the storm surge. There are some statements in this paragraph in this line that I do not think are correct (lines 434-435, 437-438, 443-444). The only exception I can think of is the arrival of tropical-like cyclones in the future climates to the European coasts. However, these would not be well captured by the coarse resolution models anyway. I think this part needs to be reconsidered.

On a personal note, I find the reading more difficult with the use of so many acronyms. My preference would be to avoid the use of at least some of them. For example: SS as storm surges, or even SDM and DDM could be referred to as, simply, statistical model and dynamical model.

**Other comments:**

- Line 92: I am unsure what this means. Perhaps that one in every N(?) coastal points are analysed? If so, what is the averaged distance among coastal points? Please, clarify.
- Figure 1 has a wrong caption. Reference to Fig 1c in line 105 is unclear.
- Lines 99-104: please, provide references here. This pattern is shown multiple times in the literature.
- Table S1: homogenise units.
- Figure S2: Please, increase the size of the figure and the font size. It is not readable.
- Line 150: SS (I guess storm surge) has not been defined. Please, limit the use of acronyms.
- Line 156: I do not see the reason to include both the gradient of SLP and the winds. At 1deg resolution, they are likely the same fields, and this would be overfitting the model. Please, discuss.
- Figure 4: units are missing in the legends.
- Lines 253-254: by errors, do you mean the uncertainties in the maximum likelihood adjustment of GEV? You also show 100-year return levels in the projections.
- Line 271: the underprediction of high storm surges is larger when trained with the hindcast. This can be due to the hindcast having smaller storm surges than the historical simulations. It would be worth checking if this is the case. That would mean that the model extrapolation is biased low. In addition, the climate models have been bias-corrected, adjusting means and variances to those in ERA5. It would also be good to check how the extremes are affected by this bias correction (probably less than the mean storm surges and this would explain these differences).
- Lines 279-280: Is this delta method necessary when the climate models are bias-corrected? I guess no for the mean characteristics of the storm surges,

- but extremes could still behave differently (also relates to my previous point above).
- Figure 6: some scales seem saturated. If this is the case, it should be explained in the text (line 289 states that changes are +/-20%).
- Line 302: what does overprediction mean here?
- Lines 326-326: I most of the Mediterranean Sea the statistical approach does not provide reliable results (see my second major comment above), which means that even if the models are consistent, the result is not robust.
- Lines 355-361: The fitting of a GEV using maximum likelihood comes with its uncertainties, that are related to the sample size and its empirical statistical distribution. The increase of uncertainties in the return levels for low-probability events is inherent to the approach, so it cannot be blamed for the decrease in the confidence of the results. The high uncertainties come from the use of a relatively short record (20 years, i.e. 20 maxima). Even with a high goodness of fit of the shape parameter, the uncertainties would increase. Therefore, please, reconsider this text.

---

## Referee Comment (RC2)

**Projections of changes in extreme storm surges for European coasts using statistical downscaling by M. Irazoqui Apecechea, A. Melet, M. Menéndez, H. Lobeto, J. B. Valle-Barroso**

**COMPREHENSIVE SUMMARY**

This manuscript investigates projected changes in extreme storm surge levels along European coasts, focusing on 10 and 100 year return levels. Extreme storm surge is defined based on daily maxima, and return levels are inferred using Extreme Value Analysis (EVA) through a Peaks-Over-Threshold (POT) approach, with an average threshold of three events per year and fitting Generalized Pareto Distributions (GPD) to exceedances.

The analysis is conducted using an ensemble of 17 CMIP6 Global Climate Models (GCMs). Return levels are estimated from 30-year periods of daily maxima and compared between a historical baseline (1995–2014) and two future horizons (2035–2064 and 2070–2100).

Daily maxima storm surges are obtained through a statistical downscaling model (SDM). The SDM is trained over the baseline period using ERA5 reanalysis data and subsequently applied to GCM CMIP6 data. The methodological choice is motivated through two experiments that compare SDM output with dynamically downscaled simulations based on a subset of four GCMs. In the first experiment, four SDMs are trained individually using each GCM's historical simulation. In the second, a single SDM is trained using ERA5, which requires bias adjustment of GCM predictors and implicitly assumes stationarity of biases but avoids the need for historical simulations from each GCM.

The authors attempt to assess whether the statistical relationships learned by the SDM remain stationary under future climate conditions and whether the SDM is able to capture climate-change-induced signals in storm surge extremes.

The SDM itself is based on multiple linear regression. The predictand is daily maximum storm surge at coastal locations, while predictors are derived from atmospheric variables including sea-level pressure, sea-level pressure gradients, and wind speed. The manuscript explores 84 predictor configurations combining different variable treatments (e.g., daily mean/min/max, quadratic wind scaling, temporal aggregation around wind maxima), spatial domains of influence (3°–12° boxes), and temporal lags (from synchronous up to two days). The final configuration is selected through cross-validation and consists of daily mean sea-level pressure, daily maximum sea-level pressure gradient, squared daily mean wind speed components, computed over a 9° spatial domain with up to two days of lag.

Using the large ensemble generated, the authors present results on inter-model agreement, uncertainty, and ensemble size dependence, offering an extensive pan-European perspective on future extreme storm surge projections.

**COMMENTARY**

I congratulate the authors for the substantial effort invested in this study. The manuscript addresses an important and timely topic and provides a valuable contribution by exploring future projections of daily maximum storm surges along European coasts using statistical downscaling applied to a large CMIP6 ensemble.

The numerical and statistical workload behind this study is impressive, and the authors make a huge effort to assess the validity of their statistical downscaling approach under climate change conditions. The pan-European scope and ensemble-based perspective are clear strengths.

That said, I believe the manuscript would benefit significantly from improvements in **structure, clarity, and methodological rigor**, particularly regarding hypothesis formulation, inference, and the separation between methods and results. In its current form, methods and results are often interwoven, making the paper difficult to follow. In addition, while many validation steps are presented, the **confidence in the final projections remains limited**, partly due to the lack of formal hypothesis testing and inference in several key analyses.

My comments below are intended to be constructive. Some are necessarily subjective or based on my interpretation; please feel free to disregard them where they are not useful or where I may have misunderstood aspects of the work.

**Detailed commentary**

*Section 1 Introduction*

**Line 29**: The reference provided supports the statement on hazards well. I recommend adding a second reference explicitly supporting the role of extreme storm surges in flood risk across Europe.

**Line 32**: The statement *"Even without changes in storm characteristics, rising mean sea levels..."* should be nuanced. Its validity depends on coastal context, such as the availability of accommodation space, sediment supply, and the presence of coastal squeeze.

**Lines 31–35**: I suggest softening the claims and acknowledging contrasting findings in the literature. For example:

Sterl et al. (2009) find no statistically significant changes in the 10,000-year return value of surge heights along the Dutch coast during the 21st century and show that higher mean sea level does not necessarily affect surge height.

Land and Mikolajewicz (2019) show that extreme sea levels in the German Bight are dominated by strong internal variability and multidecadal fluctuations rather than clear climate-change signals.

Sterl et al. (2009)  https://doi.org/10.5194/os-5-369-2009

Land and Mikolajewicz (2019) https://doi.org/10.5194/os-15-651-2019

**Line 40**: The statement on low confidence in surge projection ensembles aligns with findings for GCM and RCM wind projections, which often show larger inter-model variability than model-mean changes (e.g. wind energy studies https://doi.org/10.5194/wes-7-2373-2022).

**Lines 36–41:** In my experience, robust regional projections are also limited by hydrodynamic model uncertainty related to GCM resolution, particularly due to the

sensitivity of surge models to forcing resolution in semi-enclosed basins such as the North Sea and Mediterranean Sea.

**Paragraph starting line 42:** The discussion of statistical downscaling under climate change conditions could be strengthened by referencing earlier work demonstrating its applicability and limitations.

https://doi.org/10.1175/JCLI-D-11-00687.1
https://doi.org/10.5194/gmd-13-2109-2020

**Line 51**: The phrase *"less continuous framework"* is unclear and should be clarified.

**Line 60:** Please clarify whether you mean that SDMs can be trained directly using reanalysis data.

**Lines 60–66:** The discussion of hybrid downscaling is confusing. The manuscript applies a state-of-the-art statistical downscaling method. If hybrid downscaling is mentioned, it would help to clearly distinguish between purely data-driven statistical downscaling (this work), and hybrid approaches combining targeted numerical simulations with interpolation techniques.

**Section 2 Methods**

**About astronomical tides**, are astronomical tides explicitly modelled or removed? This should be clarified early.

**Workflow schematic**: A schematic of the full workflow (numerical model calibration/evaluation, SDM calibration/evaluation, application, and analysis) would greatly improve clarity. Figure 2 appears to serve this purpose and should be introduced earlier. Figure 1 currently presents results and could be moved later.

**Line 108**: I disagree with the statement as written. The authors should demonstrate that the hydrodynamic model robustly translates wind stress and inverse barometer effects into storm surge estimates. Otherwise, the paper should be framed more explicitly as a methodological contribution.

The hydrodynamic model is a key component of the hybrid SD framework. I recommend:

introducing the hydrodynamic methodology in the Methods section, and

moving its evaluation against observations to the Results section.

Authors are encouraged to compare hindcast surge results with other available storm surge reanalysis products.

**Data description**: A clearer and more complete description of all datasets used is missing.

**Line 92**: *"We thin coastal points by a factor of 10"* is unclear. Does this refer to reduced output resolution relative to Cid et al. (2014), or to subsampling of coastal grid points?

**Table 1**: In my experience, bias correction is required to reconcile differences between hydrodynamic models forced at 3-hourly versus 1-hourly resolution.

**Line 120**: Please explicitly state that the SDM is based on multiple linear regression.

**Line 121**: Why are daily maxima chosen rather than another temporal aggregation? Clarify the distinction between predictor and predictand. As written, this section is confusing.

If daily maxima are the target variable, it would be useful to evaluate how daily maxima from the DDM compare with GESLA observations.

**PCA details**: How are atmospheric fields standardized prior to PCA? Is the PCA weighted? Wind components may dominate variance relative to pressure variables— this should be discussed.

**Lines 130–134**: Several steps here require more detailed description for reproducibility, and some claims (e.g. that the SDM "effectively" captures certain behavior) need supporting evidence.

**Bias correction and EOFs (Line 133)**: Bias correction assumes stationarity. Did the authors verify that leading EOFs from ERA5 are spatially coherent with those from bias-corrected CMIP6 fields? Similarly, was the internal variability of the principal components examined?

**Figure 2**: Currently unreadable; resolution and clarity should be improved.

*Section 2.3*

I suggest renaming this from "Calibration" to "SDM selection".

**Multicollinearity and significance**: After applying MLR, how are predictor variables assessed for multicollinearity and statistical significance?

**Figure 3**: This figure is very insightful. Please refer to sub-panel labels rather than "top" and "bottom".

Are the metrics shown in Figure 3 averaged over the five cross-validation folds? How is over- or under-fitting avoided?

Overall, I recommend moving results currently embedded in the Methods section to the Results section.

**Selection of T5-D9-L2**: The selection appears somewhat arbitrary as differences between T5, T6, and T7 are small in several regions (I acknowlede the large effort to generate a large set of model configurations). Expanding the colorbar range or using standardized error metrics could help. Figure 4 aids interpretation, but scatter plots comparing DDM and SDM time series would further strengthen evaluation.

**Figure 4 caption**: Please explicitly refer to daily maxima values.

**Line 221**: Is the period 20 or 30 years? Please clarify.

*Section 2.4 Extreme Value Analysis*

Is POT applied to daily maxima?

What percentile corresponds to an average of three exceedances per year, and over which reference period?

Please provide more detail on the GPD fitting procedure, parameter inference, and uncertainty estimation.

*Section 3 Results*

The opening paragraph of Section 3 describes methodology and should be moved to the Methods section.

**Lines 233–235**: Please clarify the experimental design used to test the stationarity of the predictor–predictand relationship. Can MLR capture changes in variability and trends, or is something else implied?

Given the bias correction and projection in ERA5 EOF space, it is important to test whether SD-MLR errors using CMIP6 inputs are homoscedastic.

**Line 235**: The claim *"To our knowledge…"* should be moved to the Introduction and supported with relevant literature.

*Section 3.1 – SDM Evaluation*

The experimental design belongs in Methods.

If 10-year return levels are used, why not also use empirical return values?

The terminology should be consistent: use *reconstruction* for past climate and *projection* for future climate.

Training appears to occur over the same period—this is benchmarking rather than validation.

The hypotheses should be stated clearly: Both SDMs (ERA5-trained and CMIP6-trained) reproduce the distribution of daily maxima from the DDM. SDM performance is stationary across climate periods.

**Figure 5**- Showing bias and variance ratios would be informative for the first hypothesis above.

**Line 272**: Please support this statement with quantitative estimates.

**Section 3.1.2 – Future ESS Changes**

I am not convinced the 10-year return level metric robustly tests the hypothesis that SDM projections capture future change.

I recommend formal hypothesis testing:

Test whether historical and future distributions differ (e.g. KS test).

Test whether GPD shape parameters differ (e.g. Wald test or bootstrap).

**Section 3.2 – Ensemble Projections**

**Line 344**: Check parentheses and referencing style.

**Lines 341 onward**: Agreement between SD and DD appears limited in regions such as the Baltic and North Sea. This weakens the stated confidence.

This section would benefit from uncertainty and inference on return-level estimates.

**Figure 7**: Please check the baseline period stated.

**Figure 8**: Parts of the caption describe methods and should be moved accordingly.

The random sampling of models is unclear: for 17 models taken 3 at a time there are 680 combinations—why 2000 random iterations?